# Impingement of deposited water nanodroplets with coming nanoparticles: A molecular dynamics study

Liwei Sun[1], Xiaojiao Zhang[1], Linglang Zhao[1], Di Lv[1], Keyan Lin[2], Yue Leng[1], Xiuling Wang(iD)[1]*

1 School of Mechanical Engineering, Changchun Technical University of Automobile, Changchun, China,
2 Jilin Polytechnic of Water Resources and Electric Engineering, Changchun, China

* l732589404@163.com

## Abstract

Nowadays, the manipulation of water droplets has received growing interest in both academia and industry. In the current work, we aim to induce a quick sweep of deposited water droplets upon surfaces using impact nanoparticles. With the help of molecular dynamics (MD) simulations, we observe the dynamic evolution of targeted systems under different conditions. For a small value of particle's size, deposition, wrapped bounce, and separated bounce take place as a progressive increase in particle's velocity. The motion of a particle can be directly captured through obscuring water molecules. Moreover, the mechanisms underlying these different dynamic evolutions have been revealed through calculating kinetic energy, surface energy, and observing snapshots. In addition, we map two phase diagrams with respect to the dimensionless input kinetic energy ($E_{k,\,dim}$), the diameter of the nanoparticle to that of the water droplet ($\Delta$), and the intrinsic wettability of the surface ($\theta_{sur}$) to overall investigate these effects and observe all the possible outcomes. This work paves the way to understanding the progress of impingement of nanoparticles on deposited water droplets upon surfaces, which may be a good candidate for rapidly removing deposited water droplets and recovering the hydrophobicity of solid surfaces.

## 1. Introduction

In spite of wetting phenomena having been investigated for over two hundred years, researchers still keep their enthusiasm in exploring the impingement of water droplets on solid surfaces [1]. This is due to the fact that the impingement, which involves complex solid-liquid interaction, is highly relevant to practical applications, such as self-cleaning, anti-icing, ink-jet printing, biomedical devices, and so forth [2–10]. Based on the principle of energy minimization, water droplets are perfectly spherical when they are suspended in the air in order to minimize their interfacial energy. But after these droplets come into contact with solid surfaces, only hemispheric shapes

**Data availability statement:** All relevant data are within the paper and its Supporting Information files.

**Funding:** The author(s) received no specific funding for this work.

**Competing interests:** The authors have declared that no competing interests exist.

with partially wetting solid surfaces can be observed due to additional solid-liquid interfacial energy. As such, the wetting of fluid stems from a specific interfacial energy that is proportional to the number of molecules at the wetted interface [11]. Generally, the wetting feature of water droplets can be directly expressed using the intrinsic contact angle, which can be calculated by $\cos\theta_Y=(\gamma_{sv}-\gamma_{sl})/\gamma_{lg}$. Where $\gamma_{sg}$, $\gamma_{sl}$, and $\gamma_{lg}$ are interfacial tensions at solid-gas, solid-liquid, and liquid-gas interfaces, respectively. The large Young contact angle reflects a lower degree of interaction between the solid and the liquid, and vice versa [12–18].

With the rapid development of processing technology, researchers are no longer satisfied with the investigation of natural wetting phenomena of fluids. Alternatively, their attention gradually shifts to manipulate the wetting dynamics on demand [19,20]. Inspired by lotus-leaf surfaces, artificial superhydrophobic surfaces have been successfully fabricated through an accessible way of engraving surface texture [21–23]. When water droplets are deposited on superhydrophobic surfaces, they can form a perlitic shape with extremely high apparent contact angles together with very low contact angle hysteresis. Owing to their special features, the hydrophobic surface is widely used in the field of self-cleaning and anti-icing [23]. There is a composited interface, containing both solid-liquid and liquid-gas interfaces, at the bottom of the deposited layer to reduce the solid-liquid interaction. It can be used to account for the enlarging apparent contact angle. Generally, the Cassie model is adopted to quantitatively calculate the relevant apparent contact angle by considering the contribution of solid-liquid and liquid-gas composited interface [24], expressed as $\cos\theta_c=-1+\varphi_{sol\text{-}liq}(\cos\theta_Y+1)$. Where, $\varphi_{sol\text{-}liq}$ is the area fraction of the solid-liquid interface. In contrast, the interstice among surface textures can be invaded by liquid and the wetting state transitions from Cassie state to Wenzel state with significant enlargement of wetted area, referred to as WT [25]. Correspondingly, the apparent contact angle of a Wenzel droplet, $\theta_w$, is closely related to the enlarged area, which can be estimated using $\cos\theta_W=r\cos\theta_Y$. Where $r$ is the roughness factor defined as the ratio of the actual area to the projected area. For the most common scenarios, the Cassie state is in a metastable state while the Wenzel state is a configuration that minimizes the global energy, i.e., a stable state. As such, the WT can be induced when the targeted system is undergoing a wide range of external stimuli, such as electric field [26], vibration [27], impingement of droplets [28], and so forth. Generally, WTs are irreversible because of an asymmetrical energy barrier separating Cassie and Wenzel basins [29,30]. Recently, the switchable transition from the Wenzel state to the Cassie state can be achieved through a special method. For example, Zhang et al. [31] reported that reversible WTs can be induced by applying electric fields. They observed that WTs from Cassie state to Wenzel state take place after the electric-field strength exceeds some critical value, while the system can restore its initial nonwetting states when the applied field is removed. Previous investigations demonstrated that the reversible transition is only possible when hydrophobic surfaces are replaced by superhydrophobic ones [32,33]. Li et al. [34] introduced a simple and novel approach to control the wettability of solid surfaces, which can modulate the wettability between hydrophilic and superhydrophobic surfaces by applying vibration to surfaces. For

controlling the coalescence of adjacent droplets, Arbabi et al. [35] investigated the coalescence mechanism of droplets with surfactants upon solid surfaces, which reveals that the mechanism of surfactants' mass transfer varies with the wettability of substrates. Moreover, they noted that the merged process closely resembled that of freely suspended droplets when the $\theta_Y$ exceeded 90°, while the initial contact point would narrow into a linear structure at $\theta_Y < 90°$. In 2009, Boreyko and Chen first discovered a novel phenomenon, that is, a spontaneous jump occurs over the course of adjacent droplets' coalescence [36]. Following this, many works focused on this intriguing phenomenon. The coalescence jump can be significantly enhanced when surfaces are decorated with triangular prisms [37]. The interaction between the liquid bridge linking two merged droplets and the triangular substrate can be greatly advanced, which accelerates merged droplets' retraction and enhances jumping velocity [37]. Moreover, there are several other methods for promoting the jump of merged water droplets, such as imposing vibration and an electric field on targeted systems, decorating surface texture, and so forth.

Recently, Li et al. [38] proposed an approach to achieve the purpose of removing deposited nanodroplets on solid surfaces by employing incoming nanoscale ones. Although droplets can be removed successively, the strategy has an inevitable imperfection that the collision among water droplets may produce a great deal of energy dissipation, especially at the nanoscale. Therefore, the method requires a very high degree of impact velocity, while the efficiency of removing droplets is very low. Using molecular dynamics (MD) simulations, we propose a strategy that uses nanoparticles to impact deposited water droplets upon surfaces to change the surface from wetted to nonwetting through sweeping deposited ones. We would like to study the free evolution of targeted systems and draw phase diagrams to identify all the possible outcomes under different conditions (such as different surface wettability, impacting velocities, and the diameter ratio between particles and deposited droplets). This study is expected to provide a new method for removing deposited droplets effectively, which can be employed in a variety of practical applications, such as water desalination, dropwise condensation, anti-icing, and so forth.

## 2. Model and methods

The initial configuration of MD simulations contains a deposited water droplet upon a solid surface and a suspended nanoparticle, as shown in Fig 1. The simulation domain is 36 nm in both $X$- and $Y$- directions, whose value extends to 60 nm in the $Z$- direction. The periodic boundary is applied for both $X$-, $Y$-, and $Z$- directions. The water droplet consists of 10648 water molecules, and the corresponding diameter is $D_0 = 8$ nm (see Fig. 1). For process of relaxation process, the mass center of the deposited droplet is only fixed in the horizontal direction, which allows the droplet to spread on the surface. The position of the nanoparticle is well determined by fixing its mass center at (0, 0, 18). To do so, we can ensure that there is no interaction between the particle and the water droplet during the relaxation stage.

In the current work, the mW model is employed for describing water droplets at the nanoscale [39,40]. The mW model has a special advantage that can save the simulated cost by greatly reducing the number of particles [41–43]. Although the selected model is coarse-grained, the physical properties of water, including density, surface tension, radial distribution, and self-diffusion coefficient, can be better simulated. The interaction between water-Pt and Pt-Pt can be described using the Lennard-Jones 12−6 potential, expressed as:

$$U_{LJ}(r) = 4\varepsilon \left[ \left(\frac{\sigma}{r}\right)^{12} - \left(\frac{\sigma}{r}\right)^{6} \right], r < r_{cut}$$

(1)

Where $\varepsilon$ is the depth of the potential wall, $\sigma$ is the finite distance at which the inter-particle potential is zero, and $r_c$ is the cutoff distance. We select $\varepsilon_w = 0.26838$ eV and $\sigma_w = 0.23925$ nm as the inter-particle potential to describe water-water interaction, while these two values are changed to $\varepsilon_P = 0.69375$ eV and $\sigma_P = 0.247$ nm for describing the Pt-Pt interaction [44]. In MD simulations, the intrinsic wettability of materials is typically determined by an energy parameter of $\varepsilon_{wat-Pt}$. In this study,

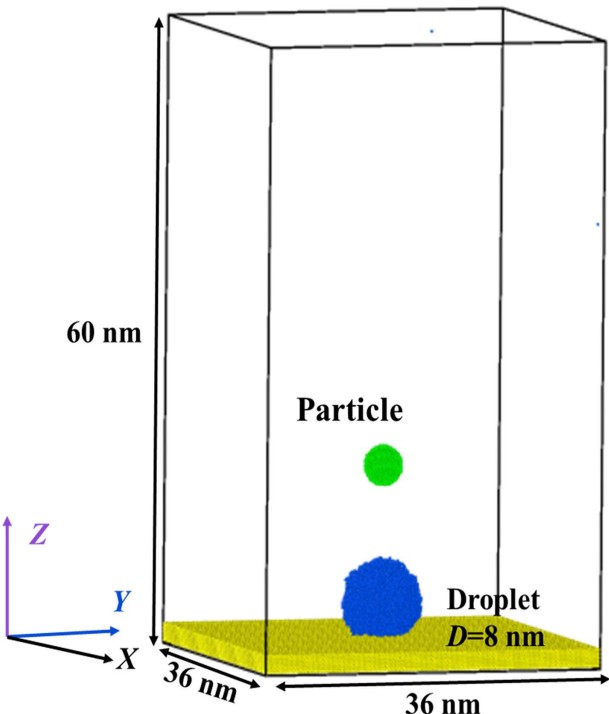

**Fig 1. Initial configuration of MD simulations containing a deposited droplet with $D_0 = 8$ nm upon a solid surface and a nanoparticle with $D_p = 2$ nm suspended in the vacuum.**

both impacting particles and the solid surface are simulated using Pt atoms. Therefore, the parameters, describing their interactions with water, are further divided into $\varepsilon_{wat\text{-}par}$ and $\varepsilon_{wat\text{-}surf}$, which indicate the interaction between water-particle and water-surface. We select different $\varepsilon_{wat\text{-}surf}$ parameters ranging from 0.0137 to 0.0021 eV to construct different wettability spanning from moderately hydrophilic to extremely superhydrophobic ($\theta_Y$ varies from 85° to 165°). For parameter of $\varepsilon_{wat\text{-}par}$, values of 0.185, 0.163, and 0.104 eV are selected to construct impact particles with $\theta_Y = 65°$, 75°, and 105°.

After we establish the initial configuration, we start to run the system in the NVT ensemble with a constant number of particles $N$, volume $V$, and temperature $T$ for 1 ns to obtain the equilibrium state. The duration is sufficiently long for the system to achieve an equilibrium state where velocity, energy, and temperature are uniformly distributed. During this period, the temperature of the water droplet is kept at 298 K using a Nose-Hoover thermostat with a temperature damping coefficient of 0.1 ps. In the second step, we use the NVE ensemble to control the total energy of the simulated system. Subsequently, we endow the incoming particle with various impacting velocities to observe dynamic features of targeted systems. The velocity-Verlet algorithm is used to update the position and velocity of each particle with a time step of 0.002 ps. The whole process is achieved using the large-scale atomic/molecular massively parallel simulator (LAMMPS) platform, and the detailed trajectories of water molecules are displayed through OVITO software [45].

## 3. Results and discussion

In this section, we first extract snapshots from MD simulations to observe the dynamic evolution of the targeted system at $V_{par} = 200$ m/s, as shown in Fig 2a. The intrinsic wettability of the surface and particle is $\theta_{Y,sur} = 135°$ and $\theta_{Y,par} = 75°$ with their parameters of $\varepsilon_{wat\text{-}surf}$ as 0.0045 eV and $\varepsilon_{wat\text{-}par} = 0.0163$ eV. The variation of energy of the targeted system is calculated and is shown in Fig 2b. Here, the energy is rewritten as a dimensionless formation, which can be obtained by dividing by the

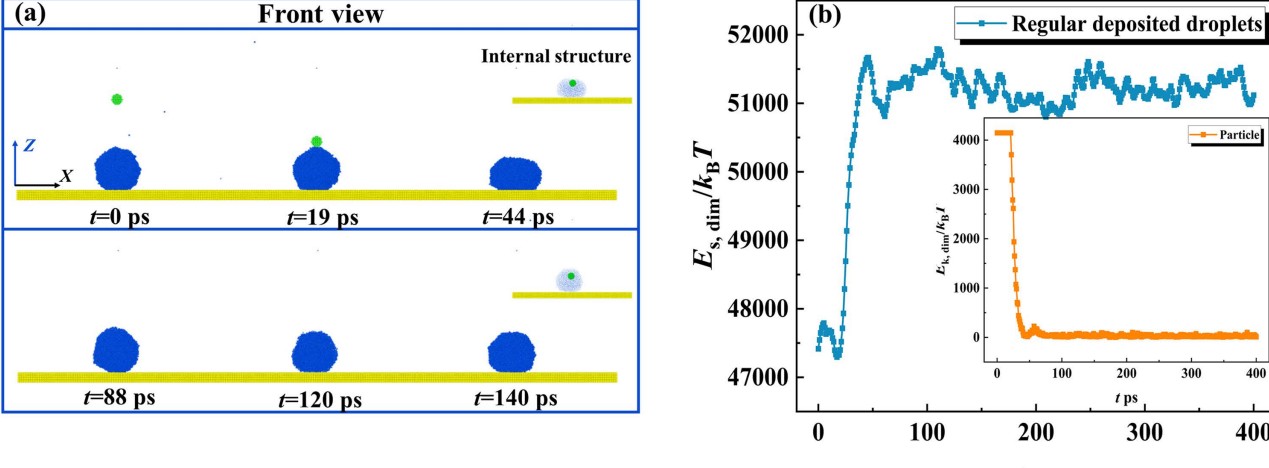

**Fig 2. Dynamic evolution of impacting particle to a deposited water droplet at the nanoscale.** The impacting velocity of nanoparticle is $V_{par}=200$ m/s, and the corresponding kinetic energy can be rewritten in a dimensionless formation as $E_{k,\,dim}=E_k/K_BT=4100$.

Boltzmann constant ($K_BT$). As shown in Fig 2a, the nanoparticle is endowed with an impact velocity and moves towards the deposited droplet. They come into contact with each other at $t=19$ ps, and the collision occurs subsequently with particle embedding in the water droplet quickly (see internal structure at $t=44$ ps). The progress of the collision is accompanied by energy conversion between the incoming particle and the water droplet. As shown in Fig 2b, the collisional process quickly reduces the particle's kinetic energy, whose value drops to null within a very short time. On the other hand, the surface energy of the droplet increases so that it spreads upon the solid surface. The fast consumption of the particle's kinetic energy traps it at the upper part of the deposited droplet, see its internal structure in Fig 2a. After $E_{s,\,dim}$ reaches its maximum value, the droplet starts to retract and vibrate with its value rising and falling. However, the energy of the droplet can not overcome the work of adhesion, and thus, the droplet eventually forms a stable wetted state after several vibrations. When velocity increases to $V_{par}=800$ m/s, the impingement of the particle can change free evolution from deposition to bounce, as shown in Fig 3a. The nanoparticle can traverse through the deposited droplet and bounce off from the solid surface, but the dynamics of the particle are limited within the water droplet during the whole impingement-bounce process, referred to as wrapped bounce. The engulfing progress occurs soon after collision begins, which generates an energy conversion following a very similar step as that for low-velocity collision, as shown in Fig 3b. Restate, there is a relative motion between the particle and the droplet, indicating that the rest particle's velocity is still higher than the reducing rate of the spreading droplet. The relative motion between the particle and water droplet endows $E_{s,\,dim}$ with rapidly attaining its maximal value (see Fig. 3b). After the particle leaves the surface, it consumes almost the remaining energy, see $t>12$ ps, see local evolution in Fig 3b. Here, the bouncing of a particle within a water droplet involves a lower rate of kinetic energy reduction compared with its falling process. To explain this, we extract the velocity distribution within a water droplet in a 2D slice at different instantaneous evolution, as shown in Figs 3c-3e. Here, the yellow-highlighted lines represent particle contours. The velocity is uniformly distributed before the internal bounce of the particle occurs, see Fig 3c. However, the internal bounce breaks this harmonious scene and produces an inverse fluid of water molecules surrounding the particle. Therefore, the internal bounce of the particle induces a complex dynamics, which produces an upward velocity in the central region while the external contour continues to spread (due to downward velocity), as shown in Figs 3d and 3e. The internal motion of the nanoparticle can raise a nanodroplet up in the vertical direction, see the fifth picture in Fig 3a. Ultimately, the wrapped bounce occurs, and the water droplet restores itself to a spherical shape in the vacuum. We next

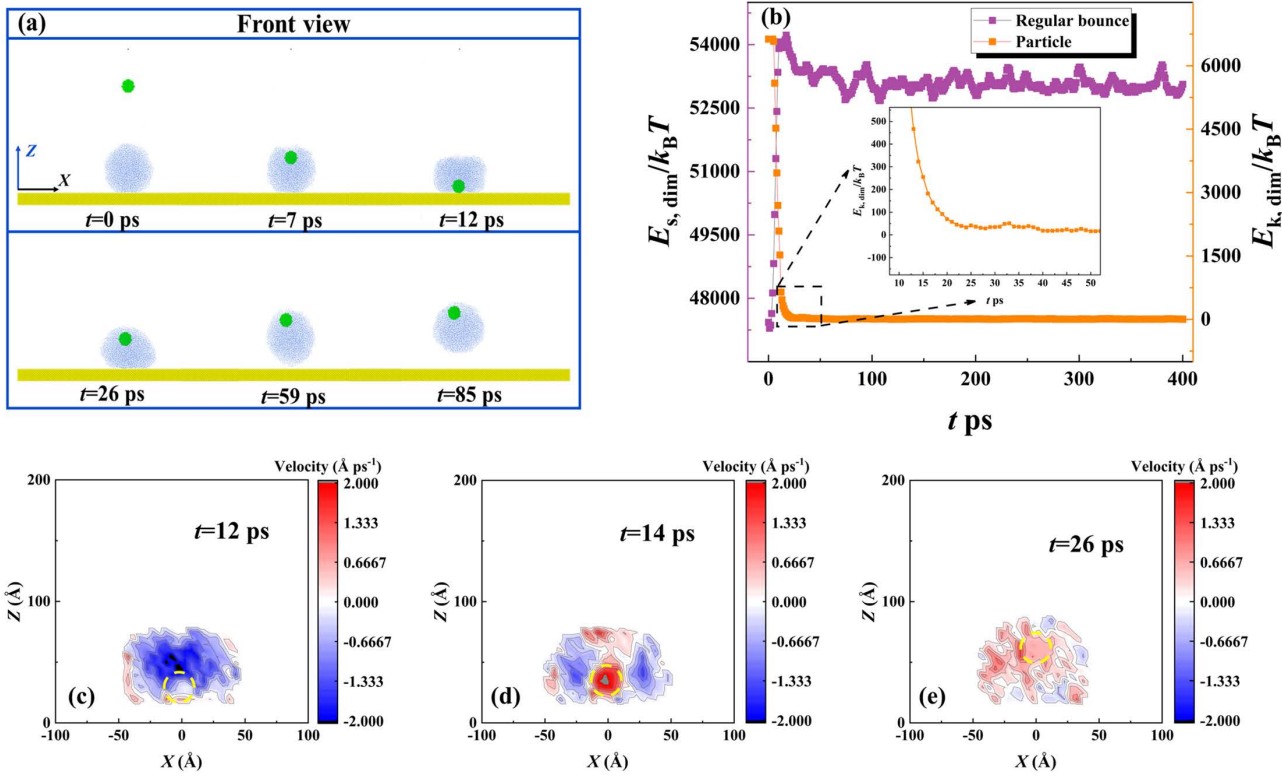

**Fig 3. Nanoparticle Impingement on Deposited Droplets: Dynamics and Energy Behavior.** (a) The dynamic evolution of targeted systems within droplet and (b) Energy conversion between kinetic energy of particle and water droplet. (c-e) Velocity distribution within water droplet at different instantaneous evolution of $t$ = 12, 14, and 26 ps.

observe the topographic evolution of bouncing droplets at different conditions with successively increasing impacting velocity, as shown in Fig 4a. A specific moment corresponding to water droplets just leaving the surface is selected as our survey objective to investigate the effect of $V_{par}$ on contact time $t_c$. Here, the velocity of the impacting particle varies from 800 m/s to 1200 m/s, and $t_c$ denotes the time from the droplet's initial contact with the solid surface until it rebounds from that surface. As $V_{par}$ increases, obvious deformation of bouncing droplets can be observed, caused by dramatic internal bounce, so that water droplets can be gradually stretched to become prolonged ones. The increasing $V_{par}$ can reduce the contact time. Moreover, the increasing $V_{par}$ to 1200 m/s involves a new progress that the nanoparticle can escape from a bouncing droplet and takes away a line of molecular cluster at the tail. We record the variation of the contact time as a function of $V_{par}$ to obtain the overall insights into how varying $V_{par}$ affects $t_c$, as shown in Fig 4b. When $V_{par}$ is below 800m/s, the bouncing droplet can not be observed because internal bouncing does not occur. On further increasing $V_{par}$, the bouncing occurs and the contact time reduces till it reaches a theoretical limit, scaled as $t_c \sim We^{1/2}Oh^{1/3}$, which is the same as that for impacting droplets. As shown in Fig 4c, when $V_{par}$ increases up to 1300 m/s, the deposited droplet can be traversed over readily by an impacting nanoparticle, which forms a tiny hole within the central part, see $t$ = 7 ps in top view. Therefore, over a large range of $V_{par}$, the nanoparticle quickly leaves from droplet at $t$ = 10 ps and raises the droplet up to form a suspension. However, the rapid detachment of the nanoparticle can not transmit too much energy, and thus, the droplet remains a ring shape and suspends in the vacuum for several tens of picoseconds, referred to as separation without bounce. Generally, the intrinsic wettability is one of the most important parameters in interfacial science, and thus, the effect of a nanoparticle's wettability has been studied. Two other intrinsic wettability of nanoparticle, $\theta_0$ = 65° and 103°, are selected as exploration objectives.

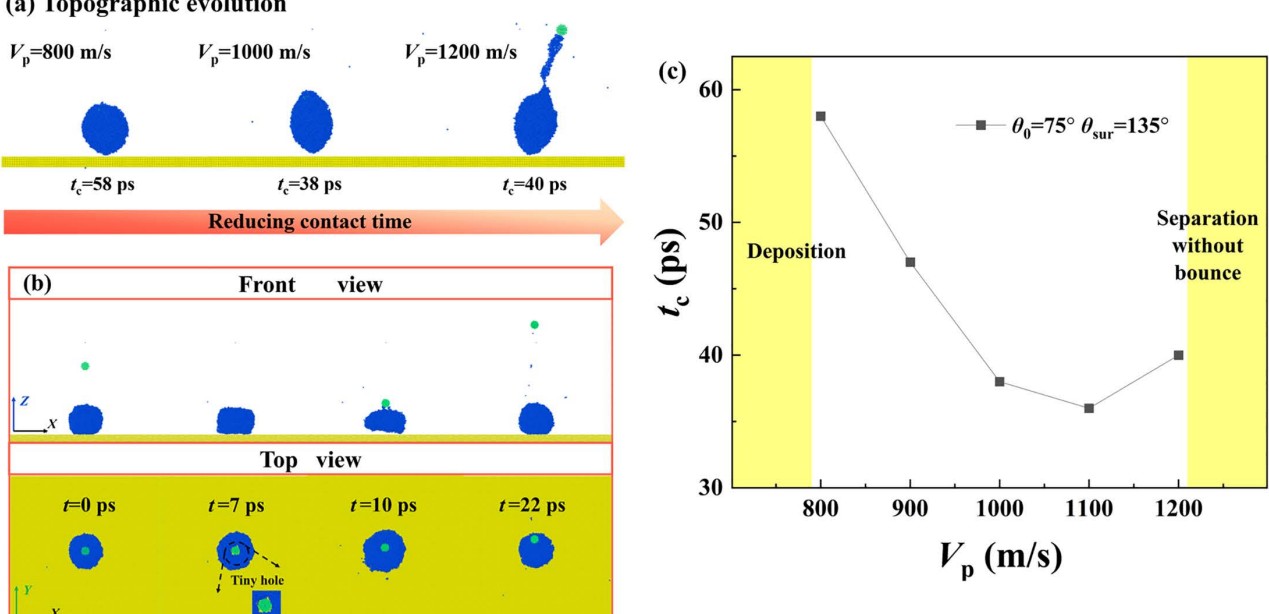

**Fig 4. Nanoparticle Impact Velocity Effects on Droplet Bouncing: Topography and Contact Time.** (a) Topological morphology of water droplets at a special moment when they just leaves off from solid surfaces at different impacting velocity. (b) Free evolution of targeted systems at an extreme velocity of 1300 m/s.

The snapshots of the free evolution of impacting systems at various conditions have been illustrated in Fig 5. We observe that during the initial impact phase-when particles remain attached to the droplet, impacting behavior over the course of the nanoparticle's impingement is independent of intrinsic wettability due to intertial-force domination. The changeable intrinsic wettability results in different solid-liquid interactions, which may produce different behavior of targeted systems. For enlarging wettability to $\theta_0 = 65°$, the increase in solid-liquid interaction leads the nanoparticle to be more adhesive to the water droplet. Therefore, the occurrence of the separated bounce can be delayed to $V_{par} = 1400$ m/s, as shown in Fig 5a. The critical value of impacting velocity for inducing separated bouncing only increases by a very small step of 100 m/s. The contact time is still around 38 ps, and its value is irrelevant to the intrinsic wettability of the nanoparticle. In contrast, when $\theta_0$ increases to 103°, the nanoparticle can easily detach from the deposited droplet at 900 m/s, leaving a deposited droplet on the solid surface (namely, separation without bounce). The phenomenon can be well understood by considering a low level of solid-liquid interaction. The kinetic energy efficiency between the particles and the surrounding water molecules is significantly reduced, thereby suppressing the bouncing behavior of the target system.

For impact particles, their size should be regarded as one of the most important parameters in altering the dynamic evolution of the targeted system. To overall investigate the size's effect, we identify the relevant outcomes at different given conditions and draw a phase diagram with respect to $E_{k, dim,}$ and $\Delta$, as shown in Fig 6. This method is widely adopted in previous investigations [46]. The value of $\Delta$ is defined as the ratio of particle diameter to droplet diameter. The selected parameters of $\Delta$ and $E_{k, dim}$ vary over a wide range from 0.25 to 1.25 for $\Delta$ and from tens $K_B T$ to 50000 $K_B T$ for $E_{k, dim}$. The parameters for the particle and solid surface are $\theta_0 = 75°$ and $\theta_{sur} = 135°$, respectively. There are a total of five outcomes in Fig 6 which are deposition, wrapped bounce, adhesion bounce, separated bounce, and separation without bounce. Apart from adhesion bounce, all the other outcomes have been introduced as mentioned above. As shown in Fig 6, different evolution takes place successively as $E_{k, dim}$ increases for a selected value of $\Delta = 0.25$. In addition, for bouncing of water

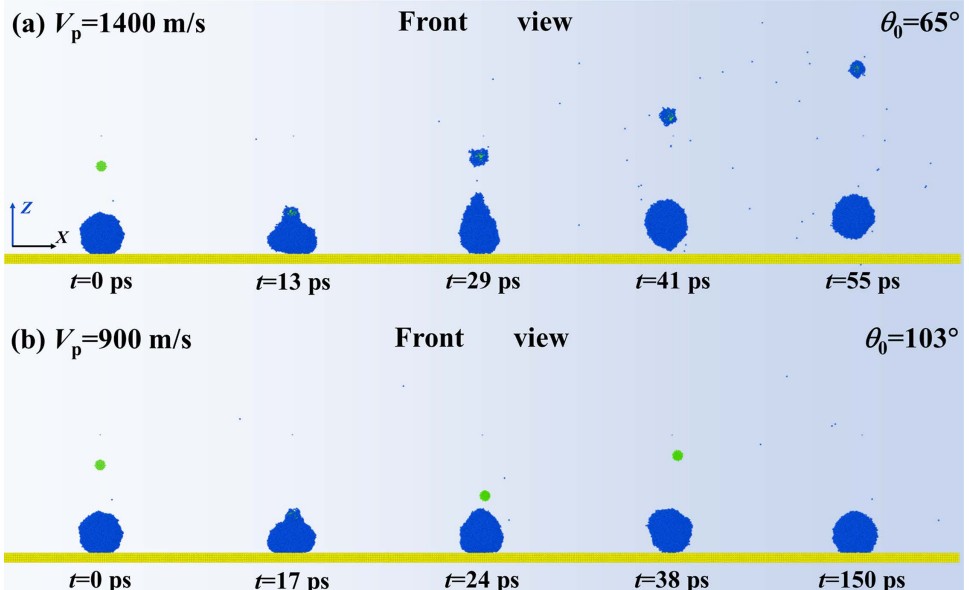

**Fig 5. Dynamic evolution of impacting systems at (a) $V_p$ = 1400 m/s and (b) 900 m/s.** The intrinsic wettability of nanoparticle is selected as $\theta_0$ = 65° and 103°, respectively.

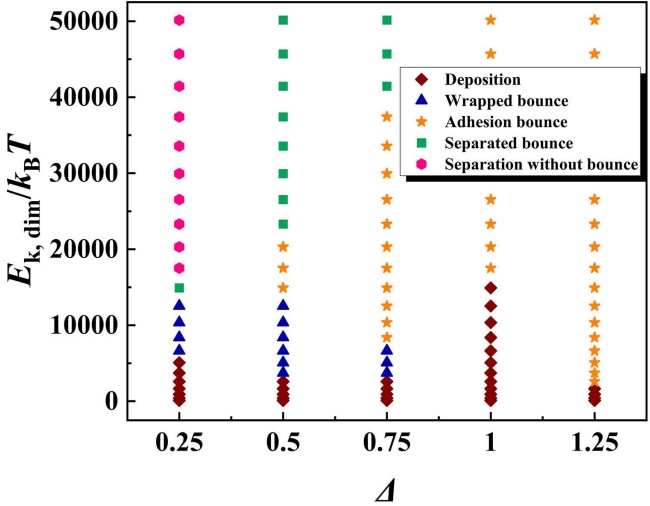

**Fig 6. Phase diagram with respect to two important parameters of $E_{k,\ dim}$ and Δ, which contains all the possible outcomes, including deposition, wrapped bounce, adhension bounce, separated bounce, and separation without bounce.**

droplets, the pattern of wrapped bounce and separated bounce only occurs at low Δ, but the adhension bounce is the main outcome at a large range of Δ (Δ = 0.75, 1, and 1.25). To have a better recognition on adhension bounce, we extract the corresponding snapshots, as illustrated in Fig 7. The snapshots show that, for Δ = 0.75 and $E_{k,\ dim}$ = 3730, the liquid initially experiences the progress of normal spreading upon the surface, and then complete wrap occurs over droplets' retraction (see Fig 7a). However, the bouncing pattern changes from wrapped bounce to adhesion bounce when Δ increases to 1.25. Here, the liquid only partly wets the bottom of the nanoparticle. Thus, the bounce of the nanoparticle can pull water

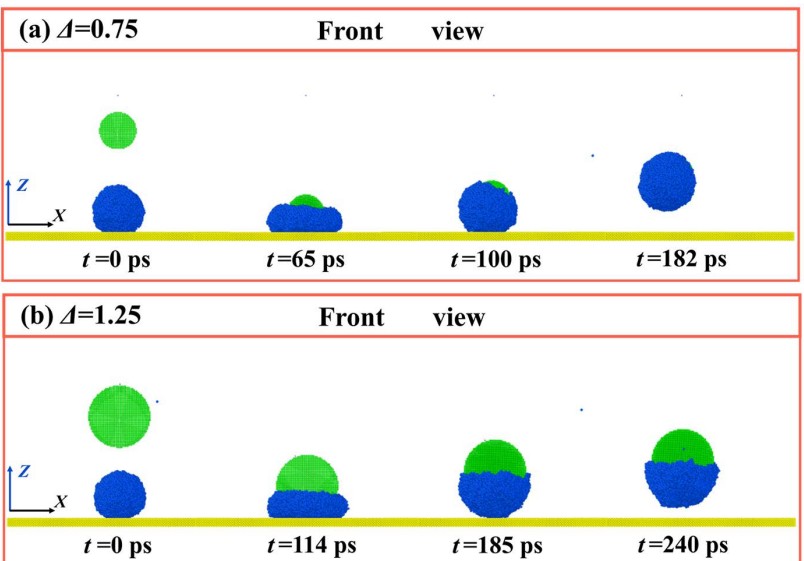

**Fig 7. Dynamic evolution of targeted systems at (a) Δ=0.75 and (b) Δ=1.25, corresponding different evolutions of wrapped bounce and adhension bounce.** Here, the value of $E_{k, dim}$ is 3730 for these two cases.

away from the solid surface under the action of strong solid-liquid interaction to form the adhesion bounce, as shown in Fig 7b. At Δ≤0.75, we observe that the wrapped bounce requires the value of Δ and $E_{k, dim}$ to be located at low range, generally for Δ≤0.75 and $E_{k, dim}$ ≤ 13000, see phase diagram in Fig 6. It indicates that the wrapped bounce is only a transitional outcome over the course of the particle's impingement. Further increasing $E_{k, dim}$, the pattern of water droplets changes from wrapped bounce to separated bounce, while the evolution alters to deposition or adhension bounce as Δ increases with a fix value of $E_{k, dim}$. The separation without bounce is a special dynamic behavior and is only possible at Δ=0.25 and a relatively high value of $E_{k, dim}$ (see Fig 6). Intriguingly, there is a special value of Δ=1 where the critical $E_{k, dim}$ required to induce bouncing behavior (including wrapped bounce and adhension bounce) is much larger than other Δ values. To obtain the reason why this phenomenon occurs, we extract the corresponding snapshots for Δ=0.75 at $E_{k, dim}$=3730, and results are illustrated in Fig 8. Snapshots show that the impingement of nanoparticle can induce water droplet to spread and retract upon the surface, see Fig 8a. The deposited water droplet can partially wrap the particle, extending to the upper part, as shown at $t$=300 ps. We next observe that the impingement of particles at increasing $E_{k, dim}$ from 103.62 to 14922, as shown in Fig 8b. We find that, as $E_{k, dim}$ increases, the water wrap the most part of the particle but with an unclosed part. Moreover, increasing $E_{k, dim}$ shows a tendency of forming a wrapped state rather than adhension, which is responsible for the increasing $E_{k, dim}$ for inducing bouncing behavior.

Ultimately, we focus on the effect of $\theta_{sur}$ on the dynamic evolution of targeted systems, and the corresponding phase diagram is shown in Fig 9a. The parameters for the particle and diameter ratio are $\theta_0$=75° and Δ=0.75, respectively. As a progressive increase in $\theta_{sur}$, one of the most prominent effects is that the critical value for inducing bouncing behavior is reduced due to the reduction of the work of adhesion. The decreasing $\theta_{sur}$ encourages separated bounce to occur while the increasing $\theta_{sur}$ is found to promote adhesion bounce, as shown in Figs 9b and 9c. The reduction of $\theta_{sur}$ makes the water droplet stretched to be an elongated shape after the impacting particle detaches from the solid surface. The bouncing water is eventually sundered to form the separated bounce, see Fig 9a. However, as $\theta_{sur}$ increases, the detachment of the nanoparticle can take away the water droplet due to the low work of adhesion, and thus, the bouncing pattern is mainly composed of adhesion bounce, as shown in Fig 9b.

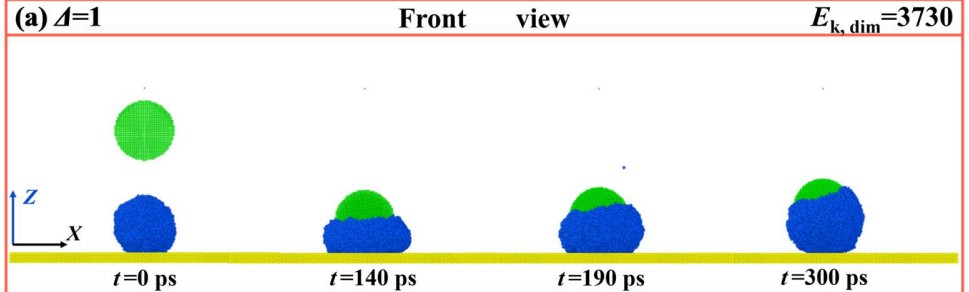

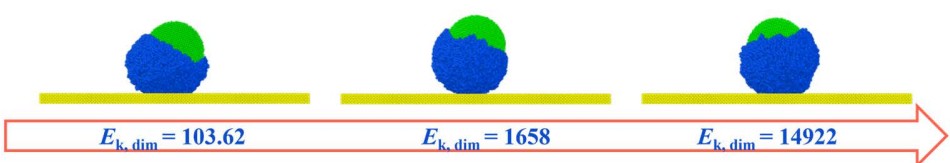

**Fig 8. Effect of Dimensionless Kinetic Energy on Nanoparticle Wrapping by Impacted Water Droplets.** (a) Dynamic evolution of impacting nanoparticle at Δ=0.75 and $E_{k,\,dim}$=3730. (b) Evolution of deposition water droplets with wrapped pattern at different given $E_{k,\,dim}$.

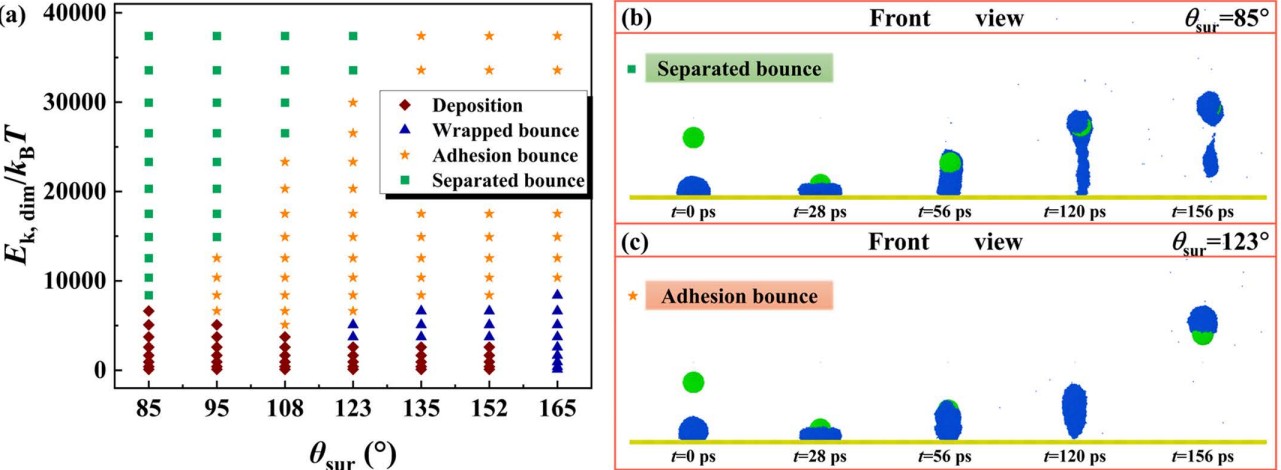

**Fig 9. $\theta_{sur}$ Effects on Dynamic Evolution and Bouncing Patterns of Targeted Systems.** (a) Phase diagram containing all the possible outcomes with $\theta_{sur}$ and $E_{k,\,dim}$. Here, the value of Δ is fixed at 0.75. (b-c) The impingement of particles on water droplet deposited on solid surfaces at $E_{k,\,dim}$=14922 but with different $\theta_{sur}$.

## 4. Conclusion

In the present work, we study the impingement of particles onto water droplets deposited on solid surfaces at the nanoscale via MD simulations. Impact processes with different particle sizes and intrinsic wettability have been simulated to reveal their effect on the dynamic evolution of targeted systems. In addition, the wettability of solid surfaces is also studied at last. The main conclusions are as follows.

The classic progress over impingement of a particle with a series of given velocities has been drawn. With a progressive increase in impact velocity, the deposition, wrapped bounce, and separation bounce take place successively. For deposition of water droplets, the energy of the particle is rapidly consumed after the particle comes into contact with the water droplet. When the particle's velocity ($V_{par}$) increases, the relative motion between the particle and water occurs, caused by the velocity difference. The internal bounce starts after it touches the basal surface. The continuous energy conversion between the kinetic energy of the particle and the surface energy of water makes it spread, retract, and even wrap bounce occur. Further increasing $V_{par}$, the particle can overcome the constraint of the droplet to form a separated state, to form a bouncing droplet/or a stable wetted one. The effect of the particle's wettability is next studied. We observe that the increasing wettability is found to promote the bouncing of water droplets, and vice versa. To obtain overall recognition of outcomes of impacting particles, we draw a phase diagram with respect to the dimensionless kinetic energy of the particle ($E_{k, dim}$) and the diameter ratio ($\Delta$) between the particle and the water droplet ($\Delta$). The total five outcomes are directly captured through observing the phase diagram, including deposition, wrapped bounce, adhesion bounce, separated bounce, and separation without bounce. The last dynamic behavior is only a transitional outcome among separated bounces. The main effect of particle size is that the small value of $\Delta$ only induces the separated bounce, while the increasing $\Delta$ changes the dynamics to the adhesion bounce. Ultimately, we explore the effect of the wettability of solid surfaces on the dynamic changes of targeted systems. As a progressive increase in $\theta_{sur}$, the pattern of bouncing water droplets changes from separated bounce to adhesion bounce, as that for the increase in $\Delta$. In addition, the increasing $\theta_{sur}$ can reduce the work of adhesion to promote the bouncing for targeted systems.

## Supporting information

**S1 File. Complete data file.**
(ZIP)

## Author contributions

**Conceptualization:** Liwei Sun, Yue Leng, Xiuling Wang.

**Data curation:** Linglang Zhao.

**Formal analysis:** Liwei Sun, Keyan Lin.

**Funding acquisition:** Linglang Zhao, Xiuling Wang.

**Investigation:** Liwei Sun, Keyan Lin, Yue Leng.

**Methodology:** Linglang Zhao.

**Project administration:** Xiaojiao Zhang.

**Resources:** Di Lv.

**Software:** Xiaojiao Zhang, Keyan Lin, Yue Leng.

**Supervision:** Di Lv.

**Validation:** Xiaojiao Zhang.

**Visualization:** Xiaojiao Zhang, Di Lv.

**Writing – original draft:** Liwei Sun.

**Writing – review & editing:** Liwei Sun, Xiuling Wang.

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
