## [Decision Letter · Decision Letter 0]

9 Dec 2025

Dear Dr. Wang,

Please submit your revised manuscript by Jan 23 2026 11:59PM. If you will need more time than this to complete your revisions, please reply to this message or contact the journal office at plosone@plos.org. Please include the following items when submitting your revised manuscript:

Kind regards,

Dr Pankaj Tomar

Academic Editor

PLOS One

Journal Requirements:

 A clean copy of the edited manuscript (uploaded as the new *manuscript* file)”.

Reviewers' comments:

Reviewer's Responses to Questions

**Comments to the Author**

1. Is the manuscript technically sound, and do the data support the conclusions?

Reviewer #1: Yes

Reviewer #2: Yes

2. Has the statistical analysis been performed appropriately and rigorously?

Reviewer #1: Yes

Reviewer #2: Yes

3. Have the authors made all data underlying the findings in their manuscript fully available?

Reviewer #1: Yes

Reviewer #2: Yes

4. Is the manuscript presented in an intelligible fashion and written in standard English?

Reviewer #1: Yes

Reviewer #2: Yes

Reviewer #1: This manuscript presents an interesting molecular dynamics study on the impingement of nanoparticles onto deposited water nanodroplets. The topic is timely and offers valuable insights for applications such as self-cleaning surfaces. I recommend publication after the authors address a few minor issues regarding presentation and technical clarification.

1.A thorough review of the spelling of words throughout the entire text is required. Specifically, there are some typographical errors and grammatical inconsistencies scattered throughout the text (e.g., "Nosadays" in the Abstract, "solid-liqud", "surface exture").

2.The authors briefly mentioned coalescence-induced droplet jumping in the Introduction and analyzed impact energies (including kinetic and surface energy) in the Results section. However, the current reference list could be enriched. There is a rich body of research concerning general droplet impact dynamics that investigates these specific behaviors and the associated energy dissipation mechanisms in detail (e.g., https://doi.org/10.1016/j.compfluid.2022.105669, https://doi.org/10.1021/acs.langmuir.2c01335, https://doi.org/10.1063/5.0047024, https://doi.org/10.1016/j.icheatmasstransfer.2024.108249, https://doi.org/10.1063/5.0118645). To further improve the completeness of the manuscript, it is suggested that the authors include a series of relevant literature, such as the examples listed above, in either the Introduction or the Results and Discussion section.

3.The authors use the dimensionless kinetic energy ($E_{k,dim}$) to analyze the results, which works well. To potentially broaden the impact of this work and make it more comparable with macroscopic fluid dynamics studies, it would be beneficial to briefly mention or estimate the corresponding Weber numbers ($We$) for the studied velocity ranges. This addition could help readers better understand the competition between inertial and capillary forces in your system.

4.The figures are informative and well-organized. For Figure 2b and Figure 3b, the insets showing the rapid energy variations are crucial for understanding the mechanism. Consider slightly enlarging these insets or increasing the font size of the axis labels in the final version to ensure they are clearly legible to the audience.

5.Please conduct a check of the bibliography for accuracy. For example, Ref [37] and Ref [40] seem to refer to the same work (Tian et al., Physics of Fluids, 2025).

Reviewer #2: Dear Editorial Team,

I am pleased to review the manuscript PONE-D-25-37158, which investigates the dynamic evolution of deposited water nanodroplets impinged by nanoparticles via molecular dynamics simulations. The research addresses a practically relevant topic with potential applications in self-cleaning, anti-icing, and surface hydrophobicity recovery, making it valuable for both academic and industrial communities.

Overall, the study is well-structured, and the MD simulation approach is appropriately employed to explore the effects of key parameters (e.g., particle velocity, size, wettability, and surface wettability). The phase diagrams summarizing five possible outcomes (deposition, wrapped bounce, adhesion bounce, separated bounce, and separation without bounce) effectively synthesize the experimental results and provide clear insights into the system’s behavior.

However, several minor revisions are recommended to enhance the manuscript’s quality and rigor:

Typos and Terminology Consistency: Correct spelling errors (e.g., "Nosadays" → "Nowadays", "swichable" → "switchable", "adhension" → "adhesion") and ensure consistent use of technical terms throughout the text.

Mechanism Elaboration: Supplement quantitative analysis of energy conversion (e.g., the ratio of kinetic energy to surface energy transformation in different dynamic processes) to deepen the understanding of underlying mechanisms.

Parameter Rationale: Clarify the justification for selecting critical parameters (e.g., the deposited water droplet diameter of 8 nm and the range of nanoparticle sizes) to strengthen the study’s methodological rigor.

Figure Annotation: Verify and correct formatting errors in figure captions (e.g., Figure 7(a) and 7(b) both label Δ=0.75, which may be a typo) to improve readability.

Comparison with Existing Work: Enhance discussions on the novelty of this study by comparing it with relevant literature, highlighting how the findings advance current knowledge in droplet manipulation.

These revisions are manageable and will significantly improve the clarity and impact of the manuscript. I recommend accepting the manuscript after minor revisions and am happy to review the revised version if needed.

Sincerely,

Reviewer

**Do you want your identity to be public for this peer review?** For information about this choice, including consent withdrawal, please see our Privacy Policy

Reviewer #1: No

Reviewer #2: No

---

## [Author Response · Author response to Decision Letter 1]

11 Dec 2025

We have upload this letter as a separate file labeled 'Response to Reviewers'.

---

## [Decision Letter · Decision Letter 1]

17 Dec 2025

Impingement of deposited water nanodroplets with coming nanoparticles: A molecular dynamics study

PONE-D-25-37158R1

Dear Author

We’re pleased to inform you that your manuscript has been judged scientifically suitable for publication and will be formally accepted for publication once it meets all outstanding technical requirements.

Kind regards,

Pankaj Tomar

Academic Editor

PLOS One

Additional Editor Comments (optional):

Reviewers' comments:

Reviewer's Responses to Questions

**Comments to the Author**

Reviewer #1: All comments have been addressed

Reviewer #2: All comments have been addressed

2. Is the manuscript technically sound, and do the data support the conclusions?

Reviewer #1: Yes

Reviewer #2: Yes

3. Has the statistical analysis been performed appropriately and rigorously?

Reviewer #1: Yes

Reviewer #2: Yes

4. Have the authors made all data underlying the findings in their manuscript fully available?

Reviewer #1: Yes

Reviewer #2: Yes

5. Is the manuscript presented in an intelligible fashion and written in standard English?

Reviewer #1: Yes

Reviewer #2: Yes

Reviewer #1: Since the authors have been well addressed the previous comments, and the current version is recommended to be published in PLOSOne

Reviewer #2: I am pleased to recommend the acceptance of this manuscript. The methodology is sound, the results are compelling, and the authors have clearly articulated the scientific significance of their findings. The work demonstrates rigorous scholarship and makes a meaningful contribution to the field of nanoscale fluid mechanics. All concerns raised in the initial review have been satisfactorily addressed, and the manuscript is now suitable for publication in PLOS ONE. I support its acceptance.

**Do you want your identity to be public for this peer review?** For information about this choice, including consent withdrawal, please see our Privacy Policy

Reviewer #1: No

Reviewer #2: No

---

## [Editor Report · Acceptance letter]

PONE-D-25-37158R1

PLOS One

Dear Dr. Wang,

I'm pleased to inform you that your manuscript has been deemed suitable for publication in PLOS One. Congratulations! Your manuscript is now being handed over to our production team.

Kind regards,

on behalf of

Dr. Pankaj Tomar

Academic Editor

PLOS One